# The role of antecedent conditions in translating precipitation events into extreme floods at catchment scale and in a large basin context

Maria Staudinger[1], Martina Kauzlaric[2], Alexandre Mas[3], Guillaume Evin[3], Benoit Hingray[3], and Daniel Viviroli[1]

[1]Department of Geography, University of Zurich, Winterthurerstrasse 190, 8057 Zurich, Switzerland
[2]Mobiliar Lab for Natural Risks, University of Bern, Bern, Switzerland
[3]Univ. Grenoble Alpes, INRAE, CNRS, IRD, Grenoble INP, IGE, Grenoble, 38000, France

**Correspondence:** Maria Staudinger (maria.staudinger@geo.uzh.ch)

**Abstract.**

In this study, we analyze how precipitation, antecedent conditions, and their spatial patterns and interactions lead to extreme floods in a large catchment. The analysis is based on 10,000 years of continuous simulations from a hydro-meteorological model chain for a large catchment, the Aare river basin, Switzerland. To account for different flood-generating processes, we

based our work on simulations with hourly time resolution. The hydro-meteorological model chain consisted of a stochastic weather generator (GWEX), a bucket-type hydrological model (HBV), and a routing system (RS Minerve), providing the hydrological basis for flood protection management in the Aare river basin.

From the long continuous simulations of runoff, snow, soil moisture and dynamic storage, we were able to assess which combinations of antecedent conditions and triggering precipitation lead to extreme floods in the sub-basins of the Aare catchment.

We found that only about 18 to 44% (depending on the sub-catchment) of annual maximum precipitation (AMP) and simulated annual maximum flood (AMF) events occurred simultaneously, highlighting the importance of antecedent conditions for the generation of large floods. For most sub-catchments in the 200-500 km$^2$ range, after return periods greater than 500 years we found only AMF caused by a triggering AMP, which is notably higher than the return periods typically used in design.

Spatial organization within a larger area is complicated. After routing the simulated runoff, we analyzed the important

patterns and drivers of extreme flooding at the outlet of the Aare river basin using a random forest. The different return period classes had distinct key predictors and showed specific spatial patterns of antecedent conditions in the sub-catchments leading to different degrees of extreme flooding. While precipitation and soil moisture conditions from almost all sub-catchments were important for more frequent floods, for rarer events only the conditions in specific sub-catchments were important. Snow conditions were important only from specific sub-catchments and for more frequent events.

## 1 Introduction

Floods in general and extreme floods in particular are a threat to infrastructure and human life. With the awareness that it is not feasible to protect everything and everybody from damage caused by floods, the goal today is to minimize damage. This is achieved by a combination of structural and technical management as well as regional development planning (e.g., hazard

maps). In this context, the return period of floods, i.e., the expected time interval between flood events exceeding a specific magnitude (Makkonen, 2006), is a crucial metric. The standard approach to flood frequency analysis is to use observations of floods and fit a theoretical extreme value distribution to estimate return periods for specific flood peaks and ultimately derive design floods for safety assessments. However, there is generally a lack of observations for extreme events such as floods, which leads to uncertain estimations of the associated return periods. The uncertainty inherent in the return periods is directly linked to the measurement accuracy and quantity of observed extreme events. For instance, established rating curves may no longer be appropriate for very large floods (Westerberg et al., 2011, 2020), leading to particularly large uncertainties in the estimation of return periods for larger floods, i.e., rarer events. Recently, there has been a growing body of literature on so-called heavy tails describing the effects of the lack of observations (Merz et al., 2022; Mathevet and Garçon, 2010; Klemeš, 2000a, b, and references therein). The issue is exacerbated because of the increasing non-stationarity of observations, which are caused by climate change but also by the impacts of human infrastructure such as water reservoirs on the streamflow and the extremes (e.g. Hingray et al., 2010).

There are some alternative approaches to overcome the lack of flood observations using historical data from archives or paleo floods (Schulte et al., 2019; Castellarin et al., 2012; Merz and Blöschl, 2008), however for ungauged catchments we face a real challenge in that we have no observations at all. For very rare flood events, approaches linking extreme precipitation to floods are often used (e.g, Naghettini et al., 1996) instead of performing flood frequency analysis based on discharge observations. Not only limited data contribute to the uncertainty of the estimated return periods, but also the choice of a suitable theoretical extreme value distribution as well as the optimization method to find appropriate parameters describing the theoretical extreme value distribution (Klemeš, 2000a, b).

To overcome the lack of data in flood frequency analysis, hydro-meteorological modelling chain approaches have been proposed and applied in a scientific context as an alternative (Lamb et al., 2016; Falter et al., 2015; Hundecha and Merz, 2012; Cameron et al., 1999; Viviroli et al., 2022). In these hydro-meteorological modelling chain approaches, meteorological scenarios from a weather generator force a hydrological model which performs continuous simulation (CS) (Beven, 1987) of discharge over long to very long periods, including floods. These approaches provide a considerably larger pool of realistic weather configurations that can potentially lead to floods compared to the pool of observed events. Thus, this approach (1) allows estimating rarer flood events and (2) provides a more robust basis for floods of medium to high return periods than is possible with observations alone. There are several additional advantages when combining CS with weather generators: it allows the exploration of a very large panel of different hydro-meteorological configurations, with different combinations of weather and catchment hydrological states (e.g. soil moisture conditions, snow pack importance and maturity, filling states of reservoirs), i.e. it also provides flood antecedent conditions without the need for explicit assumptions about them (Calver and Lamb, 1995; Pathiraja et al., 2012; Viviroli et al., 2022). It is applicable in ungauged catchments (using parameters derived from regionalization) and allows to better link flood estimation with physical processes. Furthermore, this approach treats the processes in a spatially consistent manner and captures the space–time interactions of the relevant processes (Falter et al., 2015). It offers the possibility to extract not only flood peaks but any other feature of the hydrograph, such as for instance flood volume, and thus allows for a bi- or multivariate flood frequency analysis for safety assessment (e.g., Blazkova and Beven,

2004; Brunner et al., 2016). Moreover, this type of modelling chain can be easily extended using discharge simulations as input
to flood plains (see e.g. Lamb et al., 2016) or in a further step to a damage model (see e.g., Falter et al., 2015). Finally, this approach also allows testing future scenarios under changed conditions for instance regarding climate or land use change (see e.g., Köplin et al., 2013, 2014) or regarding different regulation in the catchment.

The sources of uncertainty shift when such a hydro-meteorological modelling chain is used for flood frequency analysis. While this approach reduces the uncertainties inherent in conventional flood frequency analysis, it adds various model uncertainties. The weather generator is a statistical model that includes numerous parameters estimated with uncertainty, and which provides a simplified representation of weather dynamics in both space and time (Lafaysse et al., 2014). The weather generator is used in a specific temporal resolution and parameterization and makes use of a given underlying meteorological station density for a specific catchment. These components introduce uncertainty in the representativity of the rainfall distribution types generated that can lead to floods in a specific region. Also the hydrological model and the routing system are subject to uncertainty. Here, the main sources of uncertainty stem from model structure and parameter uncertainty. These uncertainties could be estimated by using an ensemble of simulations with different parameter sets and thus help in the decision-making process for flood safety management (Todini, 2004; Blazkova and Beven, 2004; Wood and Lettenmaier, 2008). Some studies have attempted to use multi-model approaches that attempt to represent the structural uncertainty of hydrological simulations and particularly for extremes this approach was followed by Thébault et al. (2024). When following this methodology, however, there is also the need to make specific decisions, e.g. on how many models and which models to choose, see Gupta and Govindaraju (2023). This explains why ensemble approaches are more commonly used so far, i.e. a single model is run multiple times with different input data, parametrizations, or initial conditions.

Floods can be generated by different processes, and a specific amount of rainfall trigger a flood in some cases. In other cases, the same amount may barely increase the discharge. The outcome depends on the intensity, duration, and spatial distribution and the localization of the precipitation event as well as on the antecedent conditions within the catchment.

The antecedent conditions of the catchment are shaped by its history of drying and wetting over time. These dynamics depend on catchment properties that allow for a large or only a small storage capacity, and on spatio-temporal interactions in the catchment (activated or not, see Tarasova et al. (2019) and references therein) such as how the stream network is connected during a precipitation event and other aspects of functional connectivity (Blume and van Meerveld, 2015) within a catchment. Knowledge of the relationship between antecedent catchment hydrological conditions and meteorological conditions during the event can help better estimate floods (Nied et al., 2017; Brunner et al., 2021). Many studies showed the importance of the antecedent moisture conditions for flood generation in catchments of various scales, revealing a notable influence for the streamflow response to a preceding extreme rainfall event (e.g. Michele and Salvadori, 2002; Berthet et al., 2009; Brocca et al., 2008; Bennett et al., 2018; Merz and Blöschl, 2009; Nied et al., 2013). By linking these relationships between antecedent conditions, triggering precipitation and catchment response to flood frequency, more accurate estimations can be made, such as those for inundated areas (Sikorska et al., 2015; Brunner et al., 2017).

Many flood-generating processes can be captured by daily streamflow observations(Nied et al., 2013; Falter et al., 2015). However, some flood-generating processes occur on a very short temporal scale and require hourly or even finer data resolution

to capture the potentially critical space-time dynamics within a catchment. The final flood at a specific site depends on the antecedent conditions, the triggering precipitation but also on the spatial interplay of processes occurring at different scales. Lakes and flood plains may buffer the flood peaks, while coincident floods from tributaries of different sub-catchments may increase the overall flood peak due to superposition which usually occurs at a sub-daily time scale.

**Objectives**

In this study we assess the role of antecedent conditions for floods of different return periods including extreme floods. The floods and associated antecedent conditions are extracted from very long (10'000 years) simulations from a hydro-meteorological modelling chain consisting of a stochastic weather generator optimized for the large catchment scale, a hydrological model and a routing system in hourly resolution. We specifically assess

1. the link between precipitation, antecedent conditions and return periods for the sub-catchments of the Aare river catchment and

2. their temporal and spatial interaction, accounting also for retention and confluences leading to extreme floods at the large catchment outlet.

Given the hourly time resolution and the exceptionally long precipitation time series, we anticipate to see a much greater diversity of precipitation sequences prior to floods as well as a wider variety of hydrological initial conditions in the catchment. This is expected to lead to a more robust identification of process-based relationships. Furthermore, by explicitly including hydrological routing and analyzing its effect, we can link the return periods of events to the spatial contribution of sub-catchments and the processes occurring within them.

## 2 Methods and data

### 2.1 Catchments

We studied the large-scale Aare river basin, which is one of the largest hydrological catchments in Switzerland, covering 17,700 km². It includes parts of the Alps, the Swiss Plateau and the Jura, and extends from the confluence with the Rhine at about 310 m asl to 4274 m asl in the Bernese Alps (average elevation 1050 m asl). Land use consists of pasture (36%), forest (30%), subalpine grassland (14%), bare rock (8%) and glaciers (∼2%). Streamflow is heavily managed through the regulation of the large pre-alpine lakes of Biel, Brienz, Lucerne, Murten, Neuchâtel, Thun and Zurich, and through several hydroelectric dams (Viviroli et al., 2022). The basin was divided into 127 sub-catchments, of which we selected 20 (Table 1, Figure 1) for a more detailed analysis of antecedent conditions, triggering precipitation and flood return periods. These selected sub-catchments are larger than 200 km² and vary in elevation range, slope, aspect, and hence also in hydrological regime (Table 1, Figure 1).

**Table 1.** Selection of sub-catchments of the Aare river basin with a size larger than 200 km$^2$. Average catchment response time (ACRT) in hours was calculated from the maximum cross-correlation between precipitation and discharge.

| River | Site | No. in Fig. 1 | Area | Glacier [%] | Karst [%] | Lakes [%] | Regime [1] | ACRT [h] |
|---|---|---|---|---|---|---|---|---|
| Suhre | Suhr | 1 | 243 | 0.00 | 0.02 | 5.91 | pluvial inférieur | 8 |
| Wigger | Zofingen | 2 | 366 | 0.00 | 0.00 | 0.23 | pluvial inférieur | 10 |
| Dünnern | Olten, Hammermühle | 3 | 234 | 0.00 | 30.02 | 0.00 | pluvial jurassien | 9 |
| Broye | Payerne, Caserne d'aviation | 4 | 416 | 0.00 | 0.49 | 0.00 | pluvial inférieur | 12 |
| Emme | Burgdorf, Lochbach | 5 | 661 | 0.00 | 2.14 | 0.00 | pluvial supérieur | 9 |
| Suze | Biel-Bienne, Hauserwehr | 6 | 209 | 0.00 | 69.00 | 0.00 | nivo-pluvial jurassien | 9 |
| Kleine Emme | Emmen | 7 | 478 | 0.00 | 7.04 | 0.00 | nivo-pluvial préalpin | 13 |
| Sense | Thörishaus, Sensematt | 8 | 351 | 0.00 | 8.07 | 0.13 | nivo-pluvial préalpin | 9 |
| Areuse | Boudry | 9 | 378 | 0.00 | 77.60 | 0.14 | nivo-pluvial jurassien | 17 |
| L'Orbe | Orbe, Le Chalet | 10 | 343 | 0.00 | 72.03 | 3.05 | nivo-pluvial jurassien | 14 |
| Sarner Aa | Sarnen | 11 | 269 | 0.00 | 38.72 | 3.79 | nival de transition | 49 |
| Muota | Ingenbohl | 12 | 317 | 0.00 | 35.63 | 0.95 | nival de transition | 10 |
| Kander | Hondrich | 13 | 491 | 6.69 | 27.74 | 0.26 | nivo-glaciaire | 8 |
| Simme | Latterbach | 14 | 563 | 1.75 | 22.94 | 0.00 | nival alpin | 14 |
| Engelberger Aa | Buochs, Flugplatz | 15 | 228 | 3.45 | 39.17 | 0.13 | b-glacio-nival | 10 |
| Linth | Mollis, Linthbrücke | 16 | 600 | 3.61 | 28.09 | 0.81 | nivo-glaciaire | 11 |
| Reuss | Seedorf | 17 | 833 | 8.54 | 8.93 | 0.23 | b-glacio-nival | 11 |
| Lütschine | Gsteig | 18 | 381 | 18.43 | 26.32 | 0.00 | a-glacio-nival | 6 |
| Aare | Brienzwiler | 19 | 555 | 20.54 | 12.50 | 1.08 | a-glacio-nival | 7 |
| Sarine | Broc, Château d'en bas | 20 | 636 | 0.49 | 21.24 | 0.40 | nival de transition | 12 |

[a](Weingartner and Aschwanden, 1992)

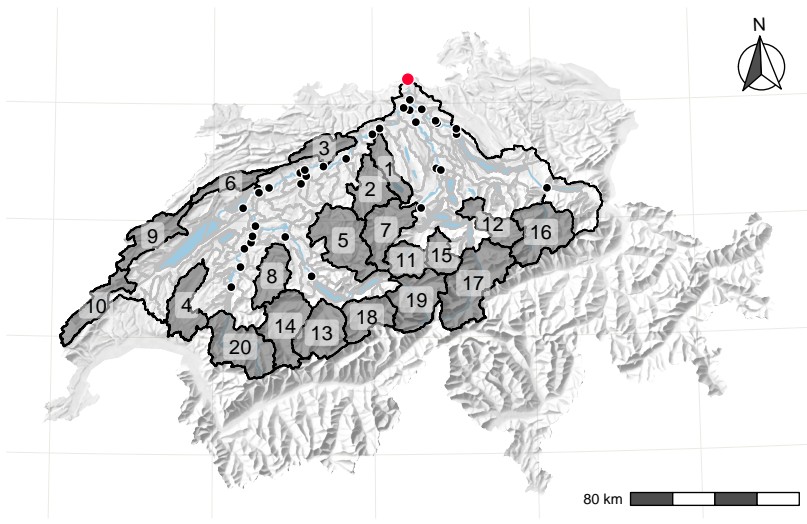

**Figure 1.** Location of the selected sub-catchments in the Aare river basin, Switzerland. The number labels are indices that can be found in Table 1. The black points are the routing system nodes that are considered in this study, and the red point is the outlet of the Aare river basin. The source of the underlying relief map is the Swiss Federal Office of Topography.

## 2.2 Hydro-meteorological modelling chain

We used the CS approach paired with a stochastic weather generator producing very long time series (here 10,000 years), which expands the pool of possible flood events and encompasses more extreme events than observations alone. This way, we
could analyze many possible but unobserved meteorological conditions causing a wide range of antecedent conditions. This enabled us to study the effect of antecedent conditions on the generation of extreme floods.

To account for many flood-generating configurations relevant in our study catchments, we based our work on simulations at hourly resolution (see average catchment response times (ACRTs) in Table1 and Charles Obled and Hingray (2009)). This also allows for a more comprehensive pool of flood events than using daily data. Moreover, the hourly time step enables a realistic
simulation and examination of the interplay of flood peaks coming from different parts of the large river catchment. Note that some runoff generation processes leading to floods that happen on finer temporal resolution are not included and some smaller scale flood processes are not covered with the structure of the hydrological model. These small-scale processes may occur in parts of most of the catchments, but their relevance diminishes as catchment area increases. For the large Aare basin, the largest events are unlikely to be governed by these processes.
The sub-catchments were selected using an appropriate discretisation level regarding the study goals, i.e., to identify large floods for the entire Aare river basin. An even finer discretisation might not lead to more insights regarding this goal, because regionally confined floods were hardly ever observed to contribute to notable floods in the Aare river basin.

### 2.2.1 Weather generator

This study exploits 10,000-year simulations of mean areal precipitation (MAP) and temperature (MAT) for each sub-catchment of the Aare river basin described in Viviroli et al. (2022). These long synthetic time series were generated by the stochastic weather generator GWEX (Evin et al., 2018, 2019), which reproduces the statistical behaviour of weather events at different temporal and spatial resolutions, focusing on extremes. GWEX is a multi-site, two-part stochastic weather generator, relying on the structure proposed by Wilks and Wilby (1999) for precipitation.

Observations of precipitation from 300 stations and observations of temperature from 77 stations were used to fit the weather generator. Precipitation and temperature records are available on a daily time step for the 1930–1980 time period and on a hourly time step for the 1981–2015 period (85 years). GWEX was fitted at the daily time step with available daily time series. GWEX was then used to generate multi-site times series of daily scenarios, further disaggregated to hourly resolution with a non-parametric disaggregation approach. For each day of the generation, the daily values are disaggregated using the spatial/temporal structure of precipitation observed for a similar day with hourly data available (Method of Fragments, see details in Viviroli et al., 2022). GWEX simulations at the different stations were finally used to calculate time series of MAP and MAT for each sub-catchment of the the Aare River.

Note that the original precipitation data were not specifically tested for stationarity. However, we used homogeneous precipitation time series. Due to the large inter-annual variability, identifying long-term trends in our precipitation data is very challenging. Isotta et al. (2019) found that seasonal precipitation trends in Switzerland are mostly statistically not significant, with weak signals of systematic change.

The weather generator was optimized for the entire Rhine river basin, rather than specifically for the Aare river basin or its individual sub-catchments. It very well reproduced the cumulative distribution functions of at-site precipitation for different temporal aggregation levels, and of all persistence properties of precipitation including the cumulative distribution functions of wet and dry spell lengths (see Evin et al., 2018). There are no known spatial inconsistencies concerning the generation of extreme events, despite in-depth evaluations as shown in Viviroli et al. (2022). Due to the hourly resolution of the disaggregated precipitation and temperature observations from meteorological stations and due to the sparse spatial coverage of the station network, very high-intensity but strongly localized events that occur at sub-hourly resolution are not reliably simulated. However, the advantage of a regionally applied weather generator is the possibility to analyze the hydrological effects emerging at the regional scale, for example, for a large hydrological catchment such as the Aare river basin.

### 2.2.2 Hydrological model

We modelled all sub-catchments within the Aare river basin using the bucket-type hydrological model HBV (Bergström et al., 1995; Seibert and Vis, 2012). The model was calibrated to observed discharge using a performance metric based on the Kling-Gupta efficiency (Gupta et al., 2009), with more weight given to the bias in the upper quantiles (50-80%) than in the classic metric, i.e. 0.25 weight to correlation between simulated and observed discharge, 0.25 weight to variability, 0.25 to bias of the

full discharge range and 0.25 weight to the upper quantiles of discharge. This was done to focus more on flood events and at the same time not to give too much weight to the uncertain peaks.

For the simulations, we forced the hydrological model with MAT and MAP from the weather generator. The generated daily temperature lapse rate was used to allocate the temperature conditions to the different elevations bands of each catchment. A constant adjustment factor of 5%/100m was applied to account for the precipitation lapse rate.

The HBV model simulates snow accumulation and melting with a degree-day approach, evapotranspiration and soil moisture storage, the drainage from the groundwater storage (response routine) and routing to finally simulate streamflow at the outlet of the catchment. In the snow routine, precipitation is adjusted for gauge under-catch and other errors if it is classified as snowfall. The extent of this correction can vary considerably depending on the hypsography of a catchment. In this study the response routine was chosen in the configuration of a non-linear drainage equation (Lindström et al., 1997), assuming an exponential increase in groundwater response with increasing water stored in the groundwater bucket. The model ultimately had 16 parameters, of which 13 were used in the calibration and 3 were fixed to values that were used in previous studies (Viviroli et al., 2022). For glaciated catchments, the model has five additional parameters, three of which are set to default values. An overview of the model parameters and limits for calibration can be found in Table 2.

For each catchment, we derived 100 plausible parameter sets using a genetic algorithm calibration procedure (Seibert, 2001), by calibrating on different nine-year-sub-periods of the available discharge records, with each sub-period resulting in 25 parameter sets. For this study, only one representative parameter set was selected from these. It is representative in the sense that it represents the median floods from the ensemble (100 parameter sets) of exceedance curves (relationships between annual maximum flood and return periods) using a percentile approach and choosing the median as proposed by Sikorska-Senoner et al. (2020). To consider the parameter uncertainty, we could use the whole ensemble or some members representing the range of possible parameter sets, but in this study we focused on the median representative parameter set and the antecedent conditions created with it.

### 2.2.3 Routing system

The simulated discharge from the HBV model was then combined and routed using the RS Minerve hydrological routing system (García Hernández et al., 2020). This system is fast to run and is well suited for application in topographically and hydraulically complex regions (regulated lakes, hydropower) such as Switzerland (Horton et al., 2022). The main impacts of bank overflow and floodplain retention were considered for a wide range of peak flows by adding channels at relevant sites, both in series and in parallel. These channels account for estimated channel flow capacity and inundated areas. Levee breaks were not considered. Stage-area-volume relationships were extracted from digital terrain information (Swisstopo, 2005) for the nine larger lakes in the Aare river basin. Six of these lakes are regulated, and the regulation rules are usually expressed as stage-discharge relationships with seasonal, monthly or even daily variations. These rules have been digitized and implemented in RS Minerve, with simplifications made where necessary. Where available and feasible, the rules were adapted for flood events (i.e., deviating from normal operation). The output nodes were placed at sites corresponding to selected gauging sites of the

**Table 2.** Parameters of the hydrological bucket type model HBV.

| Routine | Parameter | Lower limit | Upper limit | Fixed | Description |
|---|---|---|---|---|---|
| Glacier | KGmin | 0.0001 | 0.2 | | minimum outflow coefficient |
| Glacier | CFGlacier | 1 | 2 | | correction factor glacier |
| Glacier | KSI | | | 5E-05 | snow to ice conversion factor [1/h] |
| Glacier | RangeKG | | | 0 | max. minus min. outflow coefficient [1/h] |
| Glacier | CFSlope | | | 1 | correction factor slope [-] |
| Snow | TT | -2.5 | 2.5 | | threshold temperature [°C] |
| Snow | CFMAX | 0.001 | 5 | | degree day factor [mm/h °C] |
| Snow | SFCF | 0.4 | 1.6 | | snow correction factor[-] |
| Snow | CFR | | | 0.05 | refreezing coefficient |
| Snow | CWH | | | 0.1 | snow water holding capacity |
| Soil | FC | 50 | 100 | | maximum storage in soil box [mm] |
| Soil | LP | 0.3 | 1 | | threshold reduction ETP [-] |
| Soil | BETA | 1 | 5 | | shape coefficient [-] |
| Soil | PERC | 0 | 1 | | max. flow from upper to lower gw box [-] |
| Response | Alpha | 0 | 1 | | shape coefficient [-] |
| Response | K1 | 0.0001 | 0.1 | | recession coefficient (upper gw bucket) [1/h] |
| Response | K2 | 0.00001 | 0.05 | | recession coefficient (lower gw bucket) [1/h] |
| Routing | MAXBAS | 1 | 100 | | factor triangular weighting [h] |
| Precipitation redistribution | PCALT | | | 5 | lapse rate precipitation [%/100m] |

Swiss Federal Office for the Environment. For the 10,000 years simulation we assumed no changes in the current regulation and general stationarity of the system.

## 2.3 Event selection

For each of the selected sub-catchments the annual maximum flood (maximum hourly discharge), AMF, and the annual maximum precipitation sum over a fixed time window, AMP, were extracted from the full simulation period of 10,000 years. A fixed window for the precipitation sums is common in meteorological studies for extreme value statistics. However, we adjusted the fixed window size from the commonly used meteorological windows to a catchment-specific hydrologically more meaningful window, namely the average response time of the catchment. The average catchment response time (ACRT) was estimated by calculating the maximum cross-correlation between precipitation and discharge making a seasonal distinction, because of possible delays due to snow accumulation and melt in winter and spring (see Keller et al., 2018; Tarasova et al., 2019). This means that some catchments may have a fixed window of 12 hours to find the AMP, others 24 hours etc. For our set of sub-catchments, the ACRT varies from 6 to 49 hours, with a median of 11 hours. The different ACRTs may be explained to some extent by the

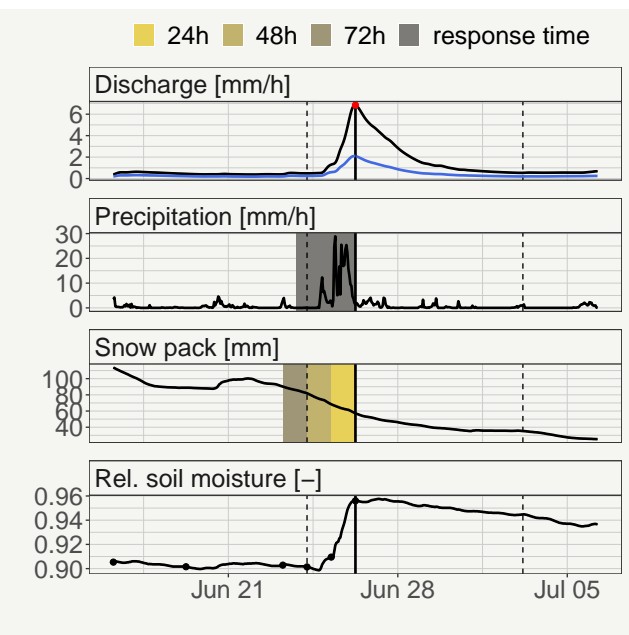

**Figure 2.** Sketch of the extracted antecedent conditions and the triggering precipitation for one exemplary flood event at hourly time resolution. The reference time is the peak of the annual mean flood (AMF), the dashed lines indicate the flood event as found by the recursive filter (Eckhardt, 2005). The grey area in the precipitation panel shows the average catchment response time (ACRT) for this example catchment. The triggering precipitation is the sum over a catchment-specific time window considering the ACRT before the beginning of the flood event. The blue line in the hydrograph (upper panel) indicates the contribution of snowmelt to the discharge. Relative soil moisture (lower panel) is extracted at different points in time (24, 48, and 72 hours before the AMF) as well as at the beginning of the flood event.

percentage of karst, river network density and the dominant runoff processes within each catchment. The flood event belonging to the AMF peak is estimated using the recursive Eckhardt filter (Eckhardt, 2005) defining the start and end of the flood event where baseflow and discharge converge. Therefore, beside the flood peak, also flood volume was estimated. The filter works well for most flood events in our catchment selection, but it tends to underestimate the flood volume for double-peak events.

Flood-triggering precipitation was defined as the precipitation that fell between ACRT before the start of the flood event
and the flood peak. Hence, any precipitation that fell before that start point in time (ACRT before the start of the AMF event) was assumed to alter the antecedent conditions but to not directly trigger the flood event. The ACRTs for each catchment are listed in Table 3. Snow has a dual function as both a temporary storage of precipitation and a delayed precipitation input in the form of snowmelt. Like rain, snow influences the antecedent conditions, and the melt water can contribute to triggering or intensifying a flood event. We excluded annual maximum precipitation events where precipitation was likely snow and not
rain, considering the substantial time gap between snow accumulation and snow melt. Rain-on-snow events are not covered by this approach and would also not be adequately simulated by the hydrological model used.

Starting from the AMF peak, we extracted a set of characteristics including simulated soil moisture, dynamic catchment storage (consisting of soil water and groundwater storage (see Staudinger et al., 2019)), snow pack at time points before the AMF and snow melt contributing to the flood as simulated by the model, as well as the associated triggering precipitation. These characteristics collectively describe the conditions during and preceding the flood. The same characteristics were also extracted before the AMP but here the discharge response to each AMP was extracted. All considered characteristics regarding flood events, antecedent conditions and triggering precipitation as well as antecedent conditions and reaction to the AMPs are listed in Table 3.

The soil moisture conditions of the catchment were calculated as relative soil moisture filling. For this the simulated soil moisture [mm] at any time was compared to the absolute maximum soil moisture that was simulated during the full simulation period [mm] in each sub-catchment. Snow conditions in the catchment were included in terms of snow pack before a flood event and in terms of relative snow melt water contribution to discharge during the flood event. The latter was calculated as the fraction of simulated snow melt in simulated discharge during the flood event.

**Table 3.** Characteristics extracted for the annual maximum flood (AMF) and the annual maximum precipitation (AMP) event to describe the antecedent conditions, triggering precipitation event and streamflow response. ACRT = average catchment response time.

| Type | AMF | AMP |
|---|---|---|
| Reference Precipitation (P) | flood peak sum of P over ACRT plus flood start to peak | max. sum of P over ACRT window |
| Discharge (Q) | flood peak flood volume flood duration fast component volume snow melt water volume | max. Q within ACRT after AMP |
| Snow conditions | snow melt in Q snow pack ACRT before | snow melt in Q snow pack ACRT before |
| Storage state | soil moisture at ACRT before | soil moisture (ACRT) |
| Seasonality | month of occurrence | month of occurrence |

## 2.4 Return period estimation

Return periods were calculated for all annual maximum flood events and all annual maximum precipitation events. We calculated the empirical return periods of each AMF and AMP based on Weibull plotting positions. The return periods were

categorized into return period classes of "10 years" (between 0 and 10 years), "100 years" (between 10 and 100 years), "300 years" (between 100 and 300 years), "500 years" (between 300 and 500 years), "1000 years" (between 500 and 1000 years), and "1000+ years" (more than 1000 years). These classes are based on the different stages of flood safety assessment and form by definition an unbalanced stratification of the full sample of annual events.

## 2.5 Occurrence of annual maxima of precipitation and flood at the sub-catchment scale

When the AMF was not caused by the AMP, we expect the greatest influence of wet catchment antecedent conditions. In these "non-matching" cases, annual maximum precipitation does not trigger the annual maximum flood and hence during and before the AMP there might be antecedent conditions that allow for the rain to be stored in the catchments and do not lead to an immediate large streamflow response. For the AMFs that were not triggered by the largest precipitation events, there are two possible cases: 1) the catchment was considerably wetter compared to the conditions during the AMP – pointing at the decisive role of antecedent catchment conditions or 2) the precipitation amount triggering the AMF was very similar to the AMP, but did not quite reach the AMP amount. In other words, in these non-matching cases a rainfall event that is slightly or markedly lower than the AMP event leads to much higher runoff production efficiency and thus ultimately to the annual maximum flood.

We considered cases as "non-matching" when the AMP did not overlap with the window of the flood event plus the preceding average response time before the flood event. Since we want to focus on hydrologically effective precipitation, we excluded AMPs where the precipitation was presumed to be snow and accumulated.

## 2.6 Critical flood conditions for the large scale catchment

From a regional management point of view, the floods that matter are those that occur at the outlet of the river basin or at a point of interest within the basin. Critical conditions at these points are formed by antecedent conditions in specific regions (spatial patterns of wetness for contributing sub-catchments), by the phase, amount, and location of precipitation, and by the combined effect of individual space-time dynamics.

By only examining many catchments individually and how precipitation and antecedent conditions shape the streamflow response at their outlets, the link to the regional importance of floods at the sub-catchment scale would be missing. For instance, if the extreme flood of one catchment is always occurring out of sync with the other sub-catchments, then there might be no great impact expected in the large catchment context. However, if two or more catchments usually exhibit strong response to precipitation inputs and their flood peaks combine and reinforce one another, then these cases become critical in the regional flood risk context.

In this study, critical floods at the large catchment scale were defined by the return periods of the floods at the outlet of the Aare river basin. In order to model the return period classes of these critical floods, we set up a classification type random forest. The spatial pattern of the antecedent conditions of the sub-catchments, the triggering precipitation within these sub-catchments, and the conditions at the critical points of the routing were used as features for this random forest.

The precipitation conditions and antecedent conditions for each sub-catchment were extracted for the individual sub-catchments trying to capture the seasonality of streamflow to travel from the outlet of each sub-catchment to the basin outlet.

The conditions of the routing system that were considered as features in the random forest were the discharge values at critical locations preceding the floods at the outlet of the Aare river basin. Again, we accounted for travel times from the outlet to each potentially relevant routing system location. The distributions of the features for the precipitation, antecedent soil moisture conditions and conditions at potentially relevant routing system locations can be found in the supplementary material.

The random forest was grown using a stratified sampling to improve the detection of the rarer return period classes given the very biased distribution of the number of flood events per return period class. The stratified sampling was set to 26, the size of flood events of the rarest class (500+). For the random forest, 5000 trees were grown and we applied 26 variables at each split, which is more than the default square root of the number of features, but as recommended in Genuer et al. (2008) better for high dimensional classification type data sets. The optimal number of trees was determined by incrementally testing different numbers of trees and evaluating the overall out-of-bag estimate of the error rate for misclassifying the return period class of flood events. We examined the variable importance of the different predictors for each flood return period class using the mean decrease in the Gini index (MDI), which measures node impurity, i.e. how well the random forest trees split the data.

## 3 Results

### 3.1 Matching and non-matching AMP and AMF - sub-catchment scale

We found that only about 18 to 44% (depending on the sub-catchment) of the annual maximum precipitation (AMP) events and annual maximum flood (AMF) events occurred simultaneously in the simulations, highlighting the importance of antecedent conditions for the generation of large floods. When looking more closely into the non-matching events, we found that numerous AMPs occurred after the AMF of that year. This means that these AMPs neither directly triggered the flood event nor contributed to wetting up the catchment prior to the flood event (Figure 3). For the rain and snow dominated catchments, 60% and more of the events of the AMP occurred after the AMF, while for the glacier influenced catchments about 40% of the AMPs occurred after the AMF. While this could be an artifact of forcing a link between AMP and AMF using blocks of years, we found that the cases with suspicious time difference in this regard were less than 2%

In addition to the general decrease of the number of events for increasing return periods, the matching and non-matching annual maxima of precipitation and discharge are not evenly distributed per return period class (Figure 4). With higher return periods there were more matching events. This indicates that increasingly extreme flood events are primarily explained by large amounts of precipitation and less by existing antecedent conditions. Nevertheless, even for very large return periods, the antecedent conditions still seemed non-negligible and for single sub-catchments even large fractions of AMF were not explained by AMP (>25% up to 75% class 1000+). The points indicating high percentage of non-matching events in the large return periods come from the Wigger river catchment (Figure 4).

Figure 5 shows the distribution of soil moisture conditions for matching and non-matching events separately and grouped for the different return period classes. Applying this separation reveals that the soils are wetter during the non-matching events than in the matching events (Figure 5). This implies that even smaller precipitation events can lead to large floods if the antecedent conditions are wetter. When comparing different return period classes, it appears that soil moisture filling increases for the more

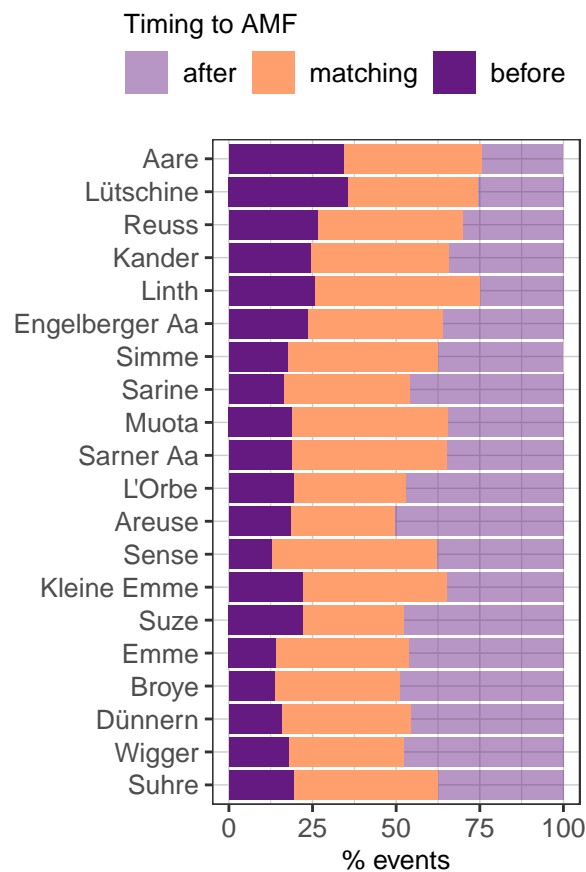

**Figure 3.** Timing of AMP to AMF events, i.e. occurrence of annual maximum precipitation (AMP) before, with or after the annual maximum flood (AMF).

extreme events (higher return periods) in the non-matching event years. In some catchments, only matching events were found in the higher return period classes, meaning that the highest precipitation triggered the highest flood. In most of the catchments studied, for a return period class of 500 years or higher, only matching events were noted. This indicates the diminishing influence of antecedent soil moisture conditions on the occurrence of rarer flood events. The Wigger river catchment stands out among the selected catchments due to the presence of non-matching events across all return period classes. In addition, only one matching event occurred for each return period class of 500 years or higher. This catchment has a specific seasonality with numerous floods driven by snow melt that result in AMF, rather than summer rainfall events.

For the glaciated catchments, the ranges of antecedent soil moisture conditions were very similar between matching and non-matching events, indicating generally more persistent wet soil moisture conditions in these catchments. This can be also be seen in the reference daily soil moisture distribution of all years. The difference in soil moisture conditions between matching and non-matching events decreases as we move from rain-dominated to snow-influenced and glaciated catchments. In rain-

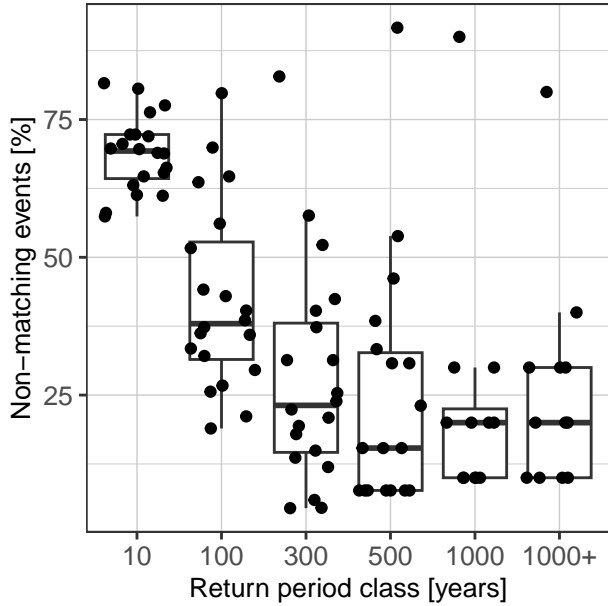

**Figure 4.** Percentage of non-matching events per return period class, excluding years with presumed snow AMPs. The higher the return period class is, the lower is the percentage of non-matching events. The dots show the percentage of non-matching events for the individual selected sub-catchments.

dominated catchments, antecedent soil moisture conditions vary widely for both matching and non-matching events. However,
there is a tendency towards wetter soil moisture conditions for the non-matching events. The ranges of soil moisture conditions tend to be narrower in the snow influenced catchments compared to the rain dominated catchments. Here, the difference between matching and non-matching events in terms of soil moisture conditions diminishes and all events occur under wetter conditions compared to the rain dominated regime type catchments. The glaciated catchments show the narrowest range and are characterized by consistently wetter antecedent soil moisture conditions for both matching and non-matching events as they
do also for the daily reference throughout the year. Generally and for all regime types, we see a more pronounced difference between the matching and non-matching as we move towards higher return periods. It is important to keep in mind that as the return period increases, the depicted density functions are constructed from a decreasing number of events. For the more frequent events, a larger pool of events is available, with a wider spread for both matching and non-matching events. Conversely, for the rarer flood events, there are only few to very few events to compare.
We analyzed the contribution of snow melt to both matching and non-matching annual maxima of precipitation and discharge by calculating the volume of simulated snow melt relative to the flood volume (Figure 6). As for the soil moisture antecedent conditions, there was more snow melt contribution in non-matching events than in matching events. In the rain dominated catchments we found a rather large difference for the snow melt contribution to the streamflow when comparing matching and non-matching events. For the snow melt dominated catchments this difference is smaller, reflecting that these catchments

**Table 4.** Confusion matrix of the classification type random forest with stratified sampling. The elements of the matrix indicate how many events found in one return period class where also modelled (prediction) into the reference return period class by the random forest.

|  |  | reference | | | | |
|---|---|---|---|---|---|---|
|  |  | 10 | 100 | 300 | 500+ | class error |
| prediction | 10 | 6278 | 1325 | 50 | 7 | 0.18 |
|  | 100 | 108 | 511 | 114 | 17 | 0.32 |
|  | 300 | 0 | 18 | 24 | 9 | 0.53 |
|  | 500+ | 0 | 4 | 6 | 16 | 0.38 |

frequently experience snow melt influenced floods. Also the Wigger catchment, which has a rain dominated regime, aligns with the description provided in the soil moisture antecedent conditions part above. It shows important snow melt contributions in the AMF up to the highest return period class. The catchments with glacier melt influenced regimes display similar distributions of snow melt contributions to floods for both matching and non-matching events.

### 3.2 Drivers and spatial patterns leading to floods at the large basin outlet

The confusion matrix of the random forest (Table 4) displays how accurately the random forest attributed the floods of a specific return period class to that same based on the provided features. For instance, in the confusion matrix (Table 4), the return period class of ten years was matched 6278 times. However, it was misclassified in the return period class of 100 years 1325 times, and in the return period class of 300 years 50 times. The random forest model appeared to have difficulty classifying the 300-year return period class, with a classification error of 52% (see Table 4), compared to the other return period classes.

The maps in Figure 7 illustrate the variable importance of the antecedent conditions regarding snow and soil moisture, triggering precipitation and routing node conditions split up for the different return period classes. The higher the mean decrease in the Gini impurity (MDI) , the higher the variable importance . From these maps, it appears that for lower return periods (100 years), soil moisture is the most important feature and for all analyzed sub-catchments. Snow pack conditions are assigned small importance in general but slightly higher for the glacier and snow influenced alpine catchments. However, triggering precipitation is only important for some sub-catchments and not important for all the glacier influenced catchments from the sub-catchment selection. The triggering precipitation of the sub-catchments Simme, Emme, and Suhre gets a slightly higher importance than the other considered sub-catchments. The MDI indicates some, but small, importance for the discharge conditions at all locations of the routing system.

Moving to higher return periods the pattern of the attribution of the individual sub-catchments is changing and precipitation generally becoming more important in explaining the attribution to the very high return period class and here also for the catchments with a glacier influenced regime. The attribution level assigned by the MDI to the triggering precipitation for the 300-year return period class is very homogeneous. While the pattern of the variable importance of soil moisture antecedent conditions remains about the same as in the lower return period class, the importance of triggering precipitation increases

compared to the soil moisture conditions when looking at the 300-year return period. Snow pack instead loses importance in classifying the return period class. For the 300-year return period class the locations in the routing system cascading downstream of the Sarine sub-catchment gain variable importance, while there was barely any assigned to these routing system locations for the return period class 100 years.

In the return period class 500+ years, the variable importance is mainly attributed to the triggering precipitation of all sub-catchments and particularly to Reuss and Muota. Also in the Broye sub-catchment, triggering precipitation becomes more prominent in assigned variable importance for these events, while it was negligible for the lower return period classes. Soil moisture does not help much in determining this return period class, with some importance being assigned to the Kander sub-catchment and some but even less to the Sense, Wigger, Suhre and Duennern sub-catchments. The Muota sub-catchment did have very little variable importance in the lower return period classes, but becomes more important in the classification of events for the 1000+ return period class.

While the contribution of snow melt to the flood at the small sub-catchment scale showed a distinct difference between the matching and the non-matching events, underscoring their importance for some events of this return period class, it did barely contribute to the flood return period classification at the river basin outlet, particularly for the rarer flood events. This can be explained by the different seasonality of floods occurring at the sub-catchment scale compared to the floods at the outlet, which are mainly summer floods. For both snow and the flow conditions at locations in the routing system there were no noteworthy variable importance assigned. Interestingly, when looking only at the change in variable importance for the conditions at the routing system locations, some changes in attribution can be seen for the different return period classes . The attribution of variable importance for the lower return period class (100 years) is at the locations closest to the Aare river basin outlet, for the higher return period class (300 years) it moves to the locations in the cascade downstream of the Sarine catchment, and for the highest return period class to the outlet of Lake Thun.

## 4  Discussion

The result that the AMP and AMF are to a large degree not occurring together are also found in other climate zones for instance in the contiguous United States Do et al. (2020) found a low correlation between changes in precipitation extremes and floods and attributed that to the small fraction of co-occurrence of these events.

As found by Nied et al. (2017) for the Elbe catchment on a large catchment scale, antecedent soil moisture conditions are important for both frequent and rare flood events observed in our study on the sub-catchments. In contrast to the finding of Nied et al. (2017), we discovered that these conditions are important up to surprisingly high return periods. The soil moisture conditions of the sub-catchments were also identified as important variables to describe the return period classes at the river basin outlet, particularly for the medium range frequent floods (100 and 300 years return period class). Only for the rarest flood events the role of the soil moisture condition could no longer be distinguished when comparing the matching and non-matching events. The decreasing effect of antecedent conditions was also found in comparisons of trends in floods and extreme precipitation events (Woldemeskel and Sharma, 2016; Tramblay et al., 2019; Bennett et al., 2018). Wasko and Nathan (2019)

found the threshold at which the importance of antecedent soil moisture conditions was negligible compared to the triggering precipitation for Australian catchments using a flood streamflow elasticity approach, already for events of around a 10 year return period.

There was a difference in the different regime types of the sub-catchments regarding the influence of soil moisture antecedent conditions. Notably, the rain dominated and slightly snow influenced catchments exhibited the most important difference in antecedent conditions between matching and non-matching events. This points at the importance of soil moisture antecedent conditions, particularly for catchments with these specific regimes. For the snow antecedent condition and its influence on the flood events, although they were to a certain degree important for the floods at the outlets of the sub-catchments, they did not emerge as important triggering factors at the large catchment scale. Also, the routing was not found to be important for modelling the return period classes of the floods at the outlet. This does not mean that they are unimportant for single events, but rather suggests that they are not as critical in classifying the events into return period classes.

Having very long CS available allowed looking at many more flood events than it would be possible with observations alone. For instance, it would not have been possible to analyze the space-time patterns that are important to describe floods on a large catchment scale with observations alone. Nied et al. (2017) used a reshuffling of meteorological and soil moisture conditions to gain more insight into the importance of hydro-meteorological processes on floods. In our approach, using the stochastic weather generator at the beginning of the hydro-meteorological modelling chain, the CS was extended even further, allowing to analyze the space-time patterns at the large catchment scale for the rare flood events as well and providing a more robust basis for the more frequent floods. Based on the CS approach, which includes a stochastic weather generator, extreme floods with return periods of more than 300 years had still relevant variable importance assigned to soil moisture conditions for almost all sub-catchments. Nevertheless, when trying to model the return period classes of the 500-year return period, the data set may have been still too small, and the floods included in the 500-year return period may have been too diverse to be properly classified with the features provided to the random forest.

## 4.1 Limitations of the study design

With our study design we could not analyze if the precipitation event was patchy or not (spatial analysis of the rain event), as for instance Tarasova et al. (2020, 2019) did within the hydrological catchments. However, we could analyze the spatial interplay of the sub-catchments of the Aare river basin with regard to large floods at its outlet. We used a lumped hydrological model for each sub-catchment with mean areal input. This means that we could not find patterns within the sub-catchments that are particularly critical for the large system. These patterns might have been informative for some flood events in specific sub-catchments, since the relationship between performance of the streamflow simulation and spatial resolution of precipitation is both scale and catchment dependent as shown for instance by Lobligeois et al. (2014) for France. However, from the regional perspective, we could analyze how the interplay of sub-catchment antecedent conditions and precipitation input as well as buffering and timing upstream the outlet of the Aare river basin influenced the floods at the outlet.

The robustness of the results depends, in part, on the type of precipitation events that is used to force the hydrological model. This assigns a crucial role to the weather generator at the beginning of the modelling chain. However, for this study the main

goal was to find conditions that lead to extreme flood events and the role of antecedent conditions therein. The way in which the weather generator was set up and optimized might not represent the full range of possible events at each sub-catchment in the region. Nevertheless, we do not expect major changes in the antecedent conditions prior to a large precipitation event, as these generally build up over a longer period of time. Note that the variability of extreme floods is inherently a multivariate variability, which as such is never well captured by multi-decadal observations. Moreover, decadal climate variability (as discussed e.g. by Vance et al., 2022) cannot be accurately represented by stochastic weather generators such as the one used in this study.

### 4.1.1 Event definition

The flood event was defined based on $Q_{peak}$ and the precipitation event was defined based on the sum of precipitation over the catchment-specific window (average response time), which does not include any information about other precipitation event properties that are potentially of interest, such as precipitation intensity or storm advancement. However, these catchment specific windows for the definition of the precipitation events are important when considering varying time scales of flood generating processes in different catchments. For Switzerland, for instance, Froidevaux et al. (2015) found a rather short discharge memory for catchments in pre-Alpine, Alpine and South Alpine regions, and that considering more than three to four days of antecedent precipitation was not relevant for flood generation. However, antecedent conditions of four or more days before the flood were found to be relevant in the Jura Mountains, in the western and eastern Swiss Plateau, and at the outlet of large lakes.

When considering return periods, a more process based event definition is not possible because the events must originate from the same population. It is not reasonable to assume that longer and shorter precipitation sums belong to the same population. This leaves us with the approach of choosing fixed time windows as basis. While this is not problematic for flood peaks, which represent a single discharge value per year, it becomes an issue for precipitation events. For instance, when precipitation events are defined over a fixed window, which is the standard approach in meteorology, we might not capture the entire precipitation event including its start and end. This can result in the loss of information about storm intensity, storm advancement, and other important factors. In addition, by looking at precipitation amounts only, we may be missing the full information on effective precipitation.

The event definition in this study relies solely on information about precipitation and discharge, and this can be done both with simulated time series (as in our hydro-meteorological modelling chain approach) and observed time series. Even if we have additional simulated variables in our approach, we could pretend not to have them and see how far we get in predicting floods using only precipitation. Precipitation events and their return periods are often used to estimate the flood return periods (Naghettini et al., 1996). Having the additional simulated variables to analyze the antecedent conditions to the flood events reveals cases where this approach is not sufficient, i.e. the cases were the annual largest precipitation event did not lead to the largest annual flood event. We found that such cases are rather common within our catchments. This challenges the assumptions made in design approaches transferring AMP to AMF on a statistical rather than at an event based basis. Hence, these results point at the important role of antecedent conditions even for relatively large return periods.

The definition of when a flood and a precipitation event match could influence the role assigned to antecedent conditions. Many AMPs occurred after the AMF and had no effect on triggering or preparing antecedent conditions. However, for AMPs before the AMFs, the definition could play a role. After conducting a sensitivity analysis by systematically altering the definition of ACRT before the flood from 0.5 to 1.5 to 2 times, we found that our findings remained unchanged (supplementary material).

### 4.1.2 Characterization of the antecedent conditions

At times, precipitation accumulates as snow, and the subsequent snow melt later in the season contributes to the liquid water input into a catchment. To accurately select the largest water input event and compare it to the largest flood response event, it becomes necessary to consider snow accumulation and melt processes. However, the buildup of the snow pack varies spatially within each sub-catchment, influenced by factors such as elevation, aspect, vegetation distribution, and wind re-distribution. This is only very roughly covered in our hydrological model by distributing precipitation into elevation zones. This spatial heterogeneity, for instance, results in varying routing times for snow melt and will probably vary strongly between sub-catchments. Moreover, the snow melting process in the hydrological model is based on a degree day approach, neglecting for instance rain-on-snow events, which can substantially contribute to the generation of floods.

A future approach to studying the antecedent conditions that lead to floods could be to look directly at snow melt and rain rather than precipitation to understand the processes and antecedent conditions that lead to floods, seasonal differences in soil moisture etc. By additionally looking at the annual maximum total water input to the system, i.e., the sum of liquid precipitation and snow melt, we might be able examine the soil moisture and other catchment storage antecedent conditions more closely. However, snow melt can be different depending on the processes involved (Sikorska-Senoner and Seibert, 2020), and in a simple snow routine, rain events that fall on a snow pack and bring energy to melt it faster would not have been included (rain-on-snow). In this study we focused on the return periods from an almost classical AMP approach versus the return periods of AMFs and found that not all AMPs necessarily lead to AMFs. One of the motivations for using this almost classical approach of a fixed window for the AMP extraction was that the AMPs usually come without further information about other possible inputs to the catchment but are derived directly from the meteorological station data.

### 4.1.3 Flood frequency analysis and flood generating processes

Hydrologically, the question can be asked whether all these maximum annual flood events can be treated as if they originated from the same population since often they are created by different flood generating processes (see e.g. Merz and Blöschl, 2008). Also from a management point of view, floods originating from different generating processes might be expected to occur more in one season than in another. In addition, they might and behave differently on flood inundation areas (Sikorska et al., 2015; Brunner et al., 2017). In the statistical analysis, floods types that dominate upper tails of the distribution may not be adequately represented, often treated as a single sample along with more frequent floods Tarasova et al. (2020). A flood type wise model was proposed by Fischer (2018); Fischer et al. (2019); Fischer and Schumann (2021), where floods from a peak-over-threshold

approach were first separated into flood types and then combined into a mixture model to calculate the return period from the joint function.

With the annual maximum flood approach, only one flood per year is analyzed regarding the antecedent conditions. If we would have chosen a peak over threshold (POT) approach instead, we could have sampled more relevant events per year. However, this approach has the downside that events could be dependent. In statistical flood frequency analysis, the assumption of event independence is however crucial, allowing these events to be treated as random variables. Moreover, employing a POT approach often involves subjective choices, such as determining the appropriate threshold (Fischer, 2018; Fischer and Schumann, 2021) and selecting which events to pool.

When we compare AMF and AMP and assess their relationship, we may come to conclusions that stem from a lack of clear differentiation regarding the size of the event. For instance, the precipitation sum preceding the AMF may be nearly as large as the AMP, or it may be much smaller. The conclusion regarding antecedent conditions could be different depending on which time window we used. Similarly, the streamflow response to an AMP might be large but not quite as large as the streamflow response that contributed to the actual AMF. Since we selected the AMP using a fixed window and adjusted the precipitation triggering the AMF based on the onset of the flood, peak and the fixed window preceding the flood onset, this comparison could not be done in a straightforward manner. The relative difference of the precipitation preceding the AMF with regard to AMP ranged, on average, from 22% to 85% across all catchments.

## 4.2 Broader impact

As outlined in the introduction, the approach of using a weather generator in a regional (large catchment) context, in combination with a hydrological model and routing, has several advantages. It implicitly "reshuffles" the initial conditions and combines them with plausible weather events for that region. This approach results in a larger and more diverse pool of flood events compared to using only observations or making assumptions about the antecedent conditions. The approach could also be applied to different large catchments within a comparable climate, dominated by similar regime types. However, one important prerequisite for generating long weather time series using a weather generator is the availability of a sufficient number of weather stations with sufficiently long records for robust estimation of plausible weather.

As in other studies comparing flood and precipitation events, such as GRADEX (e.g. Guillot and Duband, 1969; Naghettini et al., 1996), or future (Brunner et al., 2021) and past (Wilhelm et al., 2022) frequency distributions of the two variables, there are some thresholds or tipping points that emerge. These thresholds are associated with the influence of the antecedent conditions and appear to remain important even for remarkably high flood return periods in our study. This underlines their importance and emphasizes that they should not be neglected. Comparative studies applying different flood estimation methods, both event-based using statistical approaches based on streamflow data alone and continuous simulation, concluded that at sub-daily time resolution such a threshold does not occur in the event-based approaches and that these tend to underestimate floods and particularly their volume and duration for small catchments (Grimaldi et al., 2012; Rogger et al., 2012; Winter et al., 2019). Lang et al. (2014) , in their comparative study – which also included historical and paleo data, various event-based statistical approaches as well as continuous simulation – additionally emphasize that event-based approaches often lack

robustness, in particular when the available database spans only a few decades. Okoli et al. (2019) developed a framework to compare different statistical and hydrological modelling methods for estimating design floods up to 1,000 year return periods and also concluded that large differences in flood estimates can arise depending on the method chosen. However, due to the large uncertainties inherent in each method, they recommend that these methods should be used in a complementary manner in practice. Consequently, the transfer from precipitation frequency distributions to flood frequency distributions should be checked for appropriateness in each specific case.

## 5    Conclusions

In this study, we assessed the role of antecedent conditions for floods of different return periods using simulations from a hydro-meteorological modelling chain, which includes a stochastic weather generator, a hydrological model, and a routing system as basis. We focused on the relationship between precipitation, antecedent conditions and return periods for the sub-catchments of the Aare river basin. The availability of very long CS allowed the analysis of a larger number of flood events than would have been possible with observations alone. For example, it would not have been possible to analyze the space-time patterns that are crucial for describing floods on a large catchment scale using observations alone. In this way, we could investigate the temporal and spatial interactions between conditions in these sub-catchments that lead to floods at the outlet of the Aare river basin.

In the case of sub-catchments, antecedent conditions play an important role for floods with large return periods up to 500 years. This role decreases and becomes negligible only for very high return periods of more than 500 years. The regime type of the sub-catchments played a critical role: In the rain-dominated catchments, the soil moisture antecedent conditions led to the most substantial difference between matching and non-matching events of AMP and AMF. For the snow-influenced and the glacier-influenced catchments, this difference diminished.

At the large catchment scale, soil moisture antecedent conditions are critical for correctly classifying the lower return periods, but become less important as we consider higher return periods of 500 and more years. Neither snow antecedent conditions nor confluence and flow time were found to be important for classification at the outlet of the river basin when using a random forest classification type model.

Hence, it is important to check the appropriateness of transferring from precipitation frequency distributions to flood frequency distributions, as the antecedent catchment conditions are usually not negligible.

*Author contributions.*  MS, DV, BH, GE and MK conceptualized the study. AM, MK and MS developed and calibrated the models and ran the simulations. MS analyzed the data and carried out the investigation, visualized the data and drafted the paper. All authors reviewed and edited the paper.

*Competing interests.* The authors declare that they have no conflict of interest.

*Acknowledgements.* Part of this study was funded by the Swiss Federal Office for the Environment (FOEN) and the Swiss Federal Office of Energy (SFOE) in the framework of the project "Extreme Floods in Switzerland". The authors thank two anonymous reviewers for their valuable comments, which helped to improve the manuscript.

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

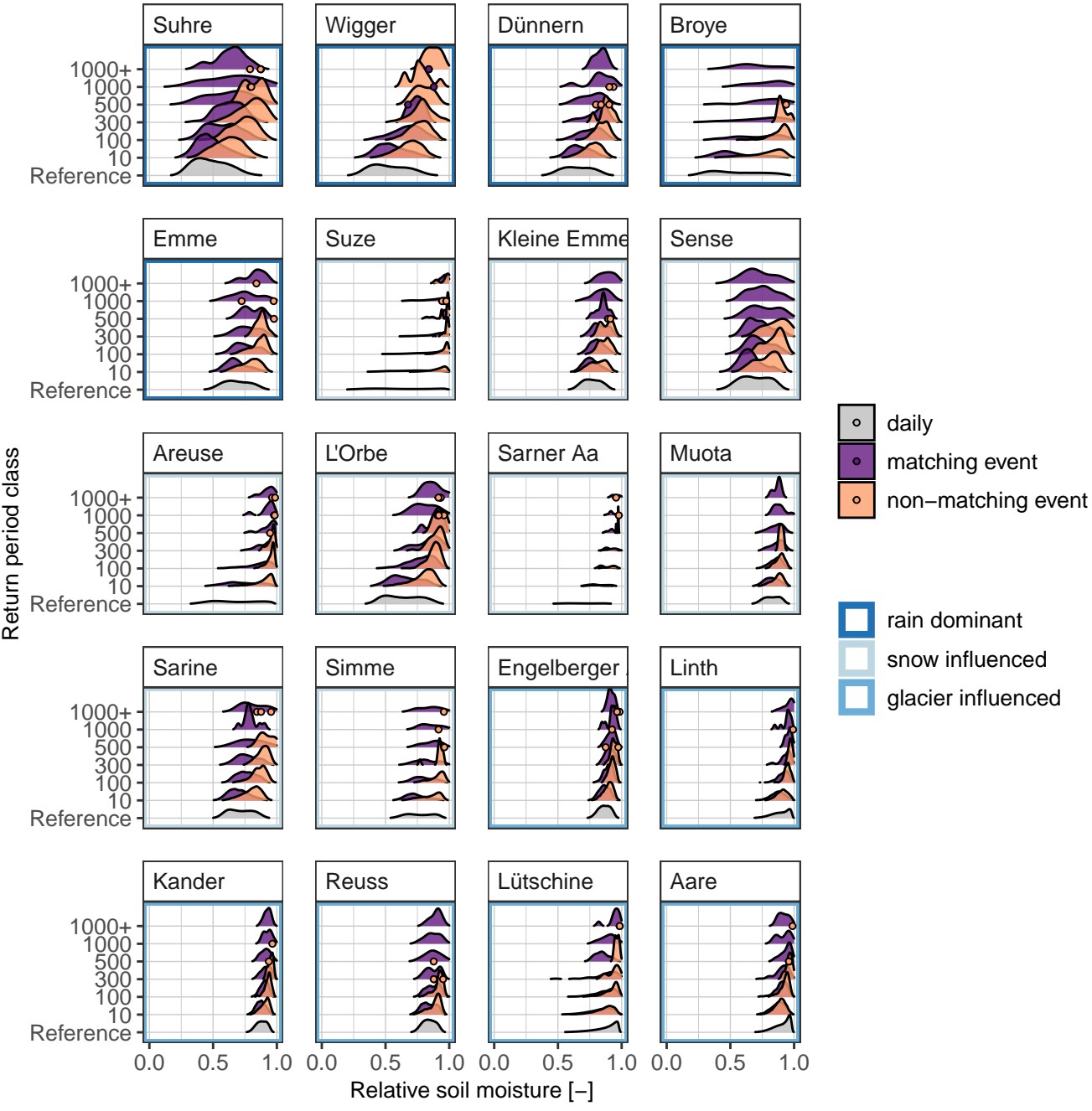

**Figure 5.** Comparison between antecedent soil moisture conditions for matching and non-matching annual maxima of precipitation and discharge and the reference of daily soil moisture conditions throughout the year (grey). In cases with two or fewer events per class, no density distribution is plotted. Instead, the soil moisture values are shown as single points. The colour frame around the panel indicates the discharge regime of the catchment.

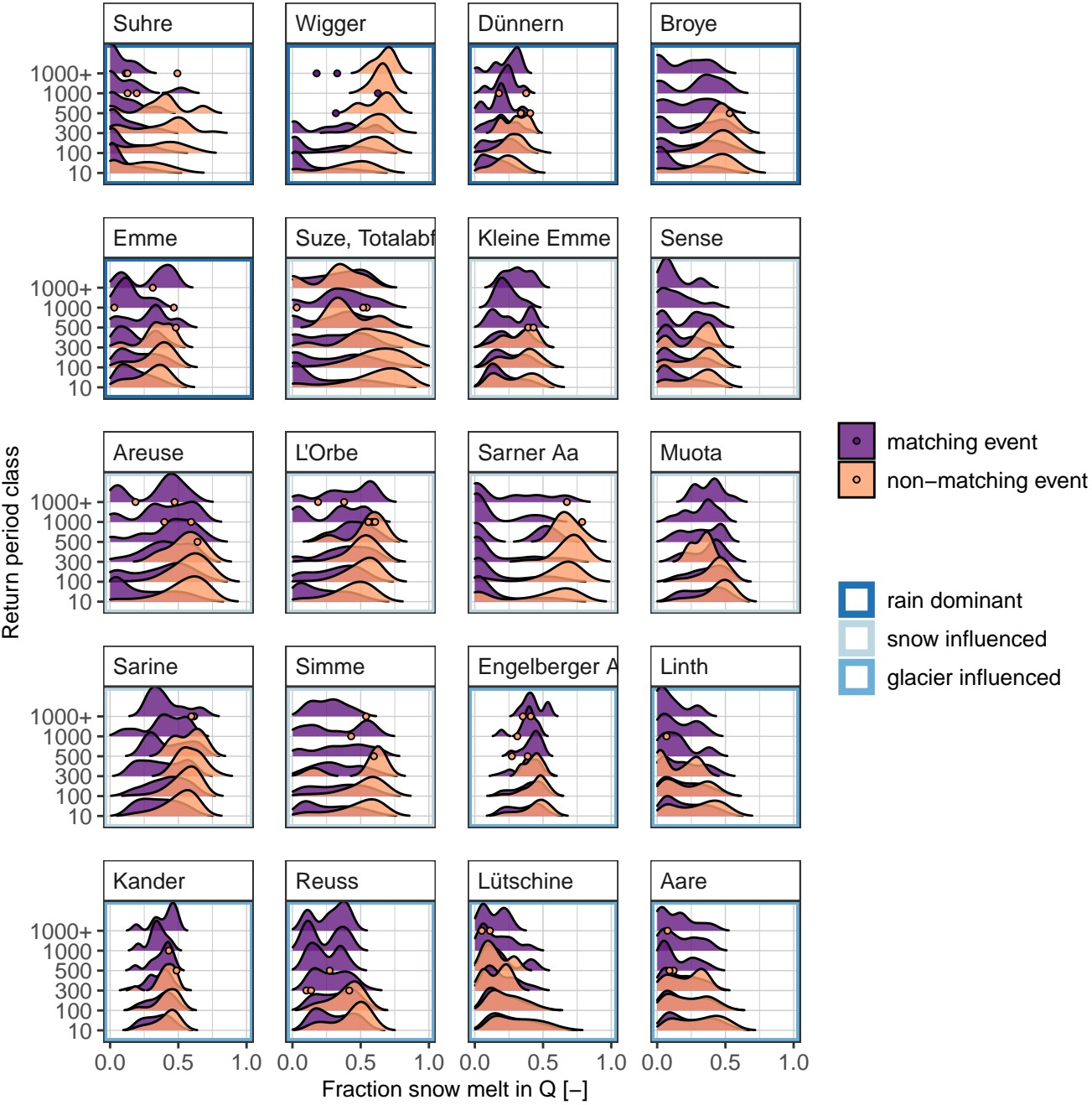

**Figure 6.** Comparison between contributing snow melt at matching and non-matching annual maxima of precipitation and discharge. In cases with two or fewer events per class, no density distribution is plotted. Instead, the contributing snow melt values are shown as single points. The colour frame around the panel indicates the discharge regime of the catchment.

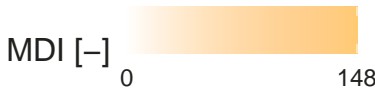

MDI [–]    0          148

100 years

PREC              SM              SNOW

300 years

PREC              SM              SNOW

500+ years

PREC              SM              SNOW

**Figure 7.** Maps of the Aare river basin and its sub-catchments showing the variable importance of triggering precipitation, soil moisture and snow conditions in the sub-catchments as well as the flow conditions at important nodes in the routing system for the return period classes 100, 300 and 500+ years. The mean decrease in the Gini impurity (MDI) is used to measure the variable importance, the darker the colour the more importance was assigned to the respective variable.