# Peer review of "The role of antecedent conditions in translating precipitation events into extreme floods at catchment scale and in a large basin context"

_EGUsphere, 2024_

## Author Comment (AC2)

*Thank you very much for your kind review. Please, find our replies below in blue and italic*

This paper discusses the connections between antecedent conditions, event rainfall and flooding in 20 large catchments in Switzerland, making use of stochastic weather generation and the HBV model.

It makes use of modelled data to overcome lack of historical flow data, which also allows the use of hourly time-series of precipitation and flow, but acknowledges the shortcomings of investigating short-duration high-intensity rainfall. Return periods of floods and rainfall were explored and quantified using random-forest classification methods, to mixed success. Interesting patterns of the lack of coherence between annual maximum flow and annual maximum event rainfall.

This paper uses one set of parameters which model the "median flood frequency curve". It would be interesting to see the more extreme flood frequency curves made use of, to see the variability of the outcomes.

*Thanks for this comment. As correctly noted, we only did the analysis for the median parameter set, which represents the "best guess" behavior for each catchment with given meteorological inputs. We would like to refrain from using other parameter sets in order to keep the focus on the impact of antecedent conditions rather than parameter uncertainty. The uncertainty aspect is the subject of ongoing research, and we feel it is beyond the scope of this paper. Insights on "more extreme" floods can be found in Viviroli et al. (2022).*

Overall this is an excellent paper, which only requires a small number of changes to finish.

*Thank you!*

Major issues:

 - A discussion of the variability of the ACRT would be useful to put some of the other values into context, as longer ACRT may have larger volumes.

*That is true. We will add other variables in context.*

 - l125. Were the rainfall and flow records tested for stationarity? Although climate change is not the focus of this paper, understanding whether the stationary stochastic generator is based on stationary data is important. Similarly, if the AMAX flow observations are not stationary, then the relationships between AMP and AMF in reality may also be changing over time.

*We did not test precipitation and streamflow records for stationarity, but we did use homogeneous precipitation series. It is difficult to identify long-term trends in precipitation and streamflow records for mean or more extreme statistics due to large interannual variability. Isotta and Frei (2019) find that seasonal precipitation trends in Switzerland are mostly not statistically significant, and signals of systematic change are weak. However, we agree that this is a strong assumption that could be discussed in more detail in the manuscript. Despite the difficulty of this stationarity issue in the context of this study, it is probably not of great importance, as we focus mainly on the effects of antecedent conditions.*

 - l125. How long were the rainfall, temperature and flow records used to extrapolate out to 10,000 year simulations? If short, then the overall distributions may under-estimate the variablity of these observations.

*The precipitation and temperature records are about 85 years long. The streamflow records were shorter for most rivers. The stochastic weather generators were trained on the 85 years of data. The hydrological models were trained on 36 years on 4 sub-periods of 9 years to derive 100 parameter sets.*

*The dispersion of the rainfall pdfs is obviously of interest. However, for flood generation, this variability is not the only one to be considered. The variability to be captured is that of composite configurations with different precipitation sequences first, and then different initial hydrological states (saturation of soils, importance and extent of snow cover, filling rates of lakes...), which the simulation method allows to explore. However, as with all models, they are based on different assumptions and mathematical representations, and it is true that some types of variability are not represented. The variability of interest for extreme floods is a multivariate variability (in a high multidimensional space), which obviously is never well observed from multidecadal period, and decadal climate variability (see, e.g., Vance et al., 2022) is not represented by the stochastic weather generators. This is a limitation that will be more clearly stated in the revised version.*

- l265. Forcing to link the AMP and AMF might lead to problems when close to the start or end of the hydrological year, leading to the pattern of "AMP after AMF". Can this be checked to see whether this is a problem?

*If we understand the comment correctly, the question is if snow accumulation hindering the direct response could link the wrong AMP to the AMF. We checked both the time between AMF and AMP as well as the likelihood of the AMP to be snow rather than rain. For the cases of snowfall during the AMP, the respective pair was removed from the analysis. The AMF and AMP with a suspicious time difference in this regard were less than 2% of the cases. We will add these aspects in the discussion.*

Minor Issues

- l97. Put numbered lists on new lines for ease of readability

*Yes, we will change to a new line there.*

- Fig1. Figure could be full page width, or blue point made bigger.

*We will adapt the color of the outlet point to make it easier to spot.*

- l126. What do you mean by "pseudo-observed"?

*Pseudo-observed means that the areal precipitation was calculated for the catchments using both hourly precipitation time series as well as daily time series, as they were available. Before calculating the hourly areal precipitation, the daily stations were disaggregated with the help of the hourly stations. Hence, pseudo as the disaggregated data were not real observations. We will change the term to not confuse.*

- Table 1. Are those standard terms for the regimes? If so, provide a reference. Otherwise define the regime terms.

*Yes, these are standard terms for the regimes in Switzerland as introduced in Weingartner & Aschwanden (1992). We will add this reference.*

- Table 4. Please add more labels to the first row and column.

*We will add that.*

***References***

*Isotta, F. A., Begert, M., & Frei, C. (2019). Long-term consistent monthly temperature and precipitation grid data sets for Switzerland over the past 150 years. Journal of Geophysical Research: Atmospheres, 124, 3783–3799. https://doi.org/10.1029/2018JD029910*

*Vance, T.R., Kiem, A.S., Jong, L.M. et al. Pacific decadal variability over the last 2000 years and implications for climatic risk. Commun Earth Environ 3, 33 (2022). https://doi.org/10.1038/s43247-022-00359-z*

*Viviroli, D., A.E. Sikorska-Senoner, G. Evin, M. Staudinger, M. Kauzlaric, J. Chardon, A.-C. Favre, B. Hingray, G. Nicolet, D. Raynaud, J. Seibert, R. Weingartner & C. Whealton (2022). Comprehensive space-time hydrometeorological simulations for estimating very rare floods at multiple sites in a large river basin. Natural Hazards and Earth System Sciences, 22(9), 2891–2920. doi:10.5194/nhess-22-2891-2022.*

*Weingartner, R., & Aschwanden, H. (1992). Abflussregimes als Grundlage zur Abschätzung von Mittelwerten des Abflusses. In Hydrologischer Atlas der Schweiz. (Vol. Tafel 5.2). Bern.*

---

## Author Response (AR1)

Dear Brunella Bonaccorso,

Thank you for your continued efforts with our manuscript. We have implemented the suggestions of both reviewers in the manuscript. Please find the updated and specific responses to the reviewers' comments below (blue, italics).

Yours sincerely on behalf of all co-authors,

Maria Staudinger

**One by one response to RC 1**

This paper discusses the connections between antecedent conditions, event rainfall and flooding in 20 large catchments in Switzerland, making use of stochastic weather generation and the HBV model.

It makes use of modelled data to overcome lack of historical flow data, which also allows the use of hourly time-series of precipitation and flow, but acknowledges the shortcomings of investigating short-duration high-intensity rainfall. Return periods of floods and rainfall were explored and quantified using random-forest classification methods, to mixed success. Interesting patterns of the lack of coherence between annual maximum flow and annual maximum event rainfall.

This paper uses one set of parameters which model the "median flood frequency curve". It would be interesting to see the more extreme flood frequency curves made use of, to see the variability of the outcomes.

*Thanks for this comment. As correctly noted, we only did the analysis for the median parameter set, which represents the "best guess" behavior for each catchment with given meteorological inputs. We would like to refrain from using other parameter sets in order to keep the focus on the impact of antecedent conditions rather than parameter uncertainty. The uncertainty aspect is the subject of ongoing research, and we feel it is beyond the scope of this paper. Insights on "more extreme" floods can be found in Viviroli et al. (2022).*

Overall this is an excellent paper, which only requires a small number of changes to finish.

*Thank you!*

Major issues:

 - A discussion of the variability of the ACRT would be useful to put some of the other values into context, as longer ACRT may have larger volumes.

*That is true. We added some statements in L213/214 and refer to the summary in Table 1.*

 - l125. Were the rainfall and flow records tested for stationarity? Although climate change is not the focus of this paper, understanding whether the stationary stochastic generator is based on stationary data is important. Similarly, if the AMAX flow observations are not stationary, then the relationships between AMP and AMF in reality may also be changing over time.

*We did not test precipitation and streamflow records for stationarity, but we did use homogeneous precipitation series. It is difficult to identify long-term trends in precipitation and streamflow records for mean or more extreme statistics due to large interannual variability. Isotta and Frei (2019) find that seasonal precipitation trends in Switzerland are mostly not statistically significant, and signals of systematic change are weak. Despite the difficulty of this stationarity issue in the context of this study, it is probably not of great importance, as we focus mainly on the effects of antecedent*

*conditions. However, we agree that this is a strong assumption that we mention in more detail in the revised version L152-155.*

- l125. How long were the rainfall, temperature and flow records used to extrapolate out to 10,000 year simulations? If short, then the overall distributions may under-estimate the variablity of these observations.

*The precipitation and temperature records are about 85 years long. The streamflow records were shorter for most rivers. The stochastic weather generators were trained on the 85 years of data. The hydrological models were trained on 36 years on 4 sub-periods of 9 years to derive 100 parameter sets. We added this information in the revised version of the manuscript (L144-146).*

*The dispersion of the rainfall pdfs is obviously of interest. However, for flood generation, this variability is not the only one to be considered. The variability to be captured is that of composite configurations with different precipitation sequences first, and then different initial hydrological states (saturation of soils, importance and extent of snow cover, filling rates of lakes...), which the simulation method allows to explore. However, as with all models, they are based on different assumptions and mathematical representations, and it is true that some types of variability are not represented. The variability of interest for extreme floods is a multivariate variability (in a high multidimensional space), which obviously is never well observed from multidecadal period, and decadal climate variability (see, e.g., Vance et al., 2022) is not represented by the stochastic weather generators. This is a limitation that is now more clearly stated in the revised version. L429-431.*

- l265. Forcing to link the AMP and AMF might lead to problems when close to the start or end of the hydrological year, leading to the pattern of "AMP after AMF". Can this be checked to see whether this is a problem?

*If we understand the comment correctly, the question is if snow accumulation hindering the direct response could link the wrong AMP to the AMF. We checked both the time between AMF and AMP as well as the likelihood of the AMP to be snow rather than rain. For the cases of snowfall during the AMP, the respective pair was removed from the analysis. The AMF and AMP with a suspicious time difference in regard of the start/end of year were less than 2% of the cases. We added this in L295/296 of the revised manuscript.*

Minor Issues

- l97. Put numbered lists on new lines for ease of readability

*Yes, we changed that (L102-106).*

- Fig1. Figure could be full page width, or blue point made bigger.

*We adapted the size and color of the outlet point to make it easier to spot.*

- l126. What do you mean by "pseudo-observed"?

*Pseudo-observed means that the areal precipitation was calculated for the catchments using both hourly precipitation time series as well as daily time series, as they were available. Before calculating the hourly areal precipitation, the daily stations were disaggregated with the help of the hourly stations. Hence, pseudo as the disaggregated data were not real observations. We rephrased to not confuse (L144ff).*

- Table 1. Are those standard terms for the regimes? If so, provide a reference. Otherwise define the regime terms.

*Yes, these are standard terms for the regimes in Switzerland as introduced in Weingartner & Aschwanden (1992). We added this reference.*

- Table 4. Please add more labels to the first row and column.

*We added these.*

**One by one response to RC 2**

The article analyses the role of antecedent moisture conditions on the estimation of extreme floods on a basin in Switzerland. The authors use a rainfall generator and a hydrological model to generate long time series on which their study is based.

I found this is an interesting paper. I have only some minor comments detailed below.

*Thank you very much!*

Detailed comments

General: Some parts of the article should be corrected by native English.

*We carefully re-read and hope to have spotted and improved these parts.*

Lines 33-35: Mathevet and Garçon (2010, https://doi.org/10.1080/02626667.2010.503934) also discussed this issue. Their analyses could be shortly commented.

*Thank you. We included it in the references (L32).*

Introduction: The following works may be interesting to cite in the context of this study :

The work by Cameron et al. (1999, https://doi.org/10.1016/S0022-1694(99)00057-8) may be worth citing here because it considered uncertainty in flood estimation using continuous hydrological modelling.

*Thank you. We included it in the references to flood frequency estimation using continuous simulation (L45).*

The work by Merz and Blöschl (2009, https://onlinelibrary.wiley.com/doi/10.1002/hyp.7168) on the controls of flood events

*Thank you. We included it in the references to flood generating processes and controls (L89).*

Lines 67-69: The authors could shortly discuss multi-model approaches in the context of flood estimation.

*Yes, there are also multi-model approaches that attempt to represent the structural uncertainty of hydrological simulations and particularly for extremes this approach was followed by Thébault et al. (2024). When following this methodology, however, we also need to decide on how many models and which to choose (see, Gupta and Govindaraju, 2023). This is why often rather ensemble approaches are used. However, it would be particularly in the context of antecedent conditions very interesting to pursue such an approach and assess the differences coming from different hydrological models in a future research indeed! We added some text on this in L72-77 of the revised version of the manuscript.*

Lines 94-99: I found the authors do not clearly show the originality of their work compared to the previous studies they cite in the introduction or others they discuss later in the article. The authors

should more clearly state the novelty of their study. What are the gaps it intends to fill compared to previous works?

*We examined processes and antecedent conditions at a finer temporal resolution than others (hourly instead of daily) and used much longer precipitation time series than has been done before, allowing for a much greater diversity of precipitation sequences prior to floods and hydrological initial conditions likely to occur in the catchment. This is expected to provide more robust results in terms of identifying process-based relationships. Furthermore, we have explicitly included hydrological routing and analyzed its effect, we have linked the return periods of events to the spatial contribution of sub-catchments and the processes within them. We have tried to make these points explicit in the revised version of the manuscript (L107-111).*

Section 2.1: For the readers who do not know the Aare basin, I found a short physical description is missing.

*We added a short physical description of the Aare basin L114-119, revised version of the manuscript.*

1: A horizontal scale and north are missing.

*We added these elements to the map.*

Table 1: The Regime column is in French.

*The regimes were actually introduced in French including all the nuances relevant for Swiss catchments, so we kept it this way and refer to Weingartner et al. (1992). For the rest and the grouping of catchments we used the three terms rain dominant, snow influenced, and glacier influenced.*

Lines 112-115: For which typical catchment size are these comments relevant?

*The small-scale flood processes are potentially occurring in parts of most of the catchments, but the larger the catchments are, the less relevant theses processes may be. For the large Aare basin, the largest events are not likely to be governed by these processes. We added this aspect (L132-134).*

Lines 121-124: Sorry but it was unclear for me whether point simulations are produced by the GWEX generator and then averaged at the catchment scale to feed the hydrological model.

*Yes, the GWEX generator first generates the weather at the meteorological stations, which are then used to calculate mean area precipitation and temperature series to the reference elevation mean catchment elevation. We added this explanation to clarify (L144-151, revised version of the manuscript).*

Line 123: The disaggregation process was unclear for me.

*We rephrased to be clearer.*

Line 133: Is there some bias in the spatial consistency of extreme events?

*To our knowledge, there are no known inconsistencies concerning the generation of extreme events, despite in-depth evaluations (Viviroli et al., 2022). We added that in the manuscript (L159-161, revised version of the manuscript)*

Line 144: At the hourly time step, the 50-80% quantiles are not very high.

*The quantiles around the median are not very high, that is correct. However, going to the higher percentiles they are higher than daily values because of the smoothing effect of aggregation. We did*

*not pick the highest percentiles on purpose to not let the model calibration be too strongly influenced by values that are increasingly uncertain themselves.*

Table 2: I did not understand why the snow correction factor (SFCF) can be below 1. Are there actually cases where measurements overestimate snow and should be corrected by SFCF below 1?

*There were no cases where this parameter was calibrated to a value less than one. Note that in the bucket-type model, each parameter is used in a conceptual way, so the parameter is intended to correct for snow, but may actually interact with other parameters and indirectly account for other processes. We kept the parameter ranges generally wider for calibration, but checked their plausibility and distribution. As noted above, the model calibration never ended in values below one for this particular parameter.*

Line 186: I was wondering whether the cross-correlation is not excessively influenced by a few extreme events. Should not the cross-correlation be calculated on transformed rainfall or streamflow (for example with square-root) to limit the influence of a few very large events?

*We calculated the cross correlation as well using a rank correlation to give less weight to a few extreme values, but the results are very similar (see Figure 1 below). The main point of using the ACRT instead of fixed windows is that we allow for catchment-specific analysis, which we believe is important and valuable in this context.*

[Figure]

*Figure 1 comparison between average catchment response time using cross-correlation with Pearson and Spearman rank correlation.*

Lines 278-279: Is this really surprising or was it expected?

*This finding is not really surprising, but it is nice that the separation of the AMF by their occurrence with the AMP or unrelated to the AMP shows this distinct pattern.*

Section 4.2: I found that the authors could further discuss the implications of their work for classical approaches of extreme flood estimation methods. I think they could discuss how their results corroborate (or not) past findings from comparative studies of flood estimation methods. I was thinking for example about the Extraflo project (Lang et al., 2014, https://doi.org/10.1051/lhb/2014010) in which a large range of flood estimation methods were compared, in gauged and ungauged conditions, using statistical approaches or methods based on continuous or event-based hydrological modelling. Other studies also attempted to compare methods (e.g. Okoli et al., 2019, https://doi.org/10.2166/nh.2019.188, and references therein).

*Thank you for these valuable references, we added them in the discussion (L519-529) in the revised version of the manuscript.*

**References**

*Gupta, A., & Govindaraju, R. S. (2023). Uncertainty quantification in watershed hydrology: Which method to use?. Journal of Hydrology, 616, 128749.*

*Isotta, F. A., Begert, M., & Frei, C. (2019). Long-term consistent monthly temperature and precipitation grid data sets for Switzerland over the past 150 years. Journal of Geophysical Research: Atmospheres, 124, 3783–3799. https://doi.org/10.1029/2018JD029910*

*Thébault, C., Perrin, C., Andréassian, V., Thirel, G., Legrand, S., & Delaigue, O. (2024). Multi-model approach in a variable spatial framework for streamflow simulation. Hydrology and Earth System Sciences, 28(7), 1539-1566.*

*Viviroli, D., A.E. Sikorska-Senoner, G. Evin, M. Staudinger, M. Kauzlaric, J. Chardon, A.-C. Favre, B. Hingray, G. Nicolet, D. Raynaud, J. Seibert, R. Weingartner & C. Whealton (2022). Comprehensive space-time hydrometeorological simulations for estimating very rare floods at multiple sites in a large river basin. Natural Hazards and Earth System Sciences, 22(9), 2891–2920. doi:10.5194/nhess-22-2891-2022.*

*Vance, T.R., Kiem, A.S., Jong, L.M. et al. Pacific decadal variability over the last 2000 years and implications for climatic risk. Commun Earth Environ 3, 33 (2022). https://doi.org/10.1038/s43247-022-00359-z*

*Weingartner, R., & Aschwanden, H. (1992). Abflussregimes als Grundlage zur Abschätzung von Mittelwerten des Abflusses. In Hydrologischer Atlas der Schweiz. (Vol. Tafel 5.2). Bern.*

---

## Author Response (AR2)

Dear Brunella Bonaccorso,

Thank you for your continued efforts with our manuscript! We have implemented the editorial corrections as suggested by reviewer #2 in the manuscript.

Yours sincerely on behalf of all co-authors,

Maria Staudinger